# Multiple Mycotoxins in Kenyan Rice

**DOI:** 10.3390/toxins13030203

**Published:** 2021-03-11

**Authors:** Samuel K. Mutiga, J. Musembi Mutuku, Vincent Koskei, James Kamau Gitau, Fredrick Ng’ang’a, Joyce Musyoka, George N. Chemining’wa, Rosemary Murori

**Affiliations:** 1Biosciences Eastern and Central Africa, International Livestock Research Institute (BecA-ILRI) Hub, P.O. BOX 30709, 00100 Nairobi, Kenya; josiah.mutuku@wave-center.org (J.M.M.); F.Nganga@cgiar.org (F.N.); J.Musyoka@cgiar.org (J.M.); 2Department of Entomology & Plant Pathology, University of Arkansas, Fayetteville, AR 72701, USA; 3National Irrigation Authority (NIA), P.O. Box 210, 10303 Wang’uru, Kenya; vkipngetich@irrigation.go.ke; 4Department of Plant Science and Crop Protection, University of Nairobi, P.O. BOX 29053, 00625 Nairobi, Kenya; jaimesgitau@gmail.com (J.K.G.); umchemin@gmail.com (G.N.C.); 5International Rice Research Institute, Eastern and Southern African Region Office, P.O. BOX 30709, 00100 Nairobi, Kenya

**Keywords:** co-contamination, food safety, multiple mycotoxins, rice, sub-Saharan Africa

## Abstract

Multiple mycotoxins were tested in milled rice samples (*n* = 200) from traders at different milling points within the Mwea Irrigation Scheme in Kenya. Traders provided the names of the cultivar, village where paddy was cultivated, sampling locality, miller, and month of paddy harvest between 2018 and 2019. Aflatoxin, citrinin, fumonisin, ochratoxin A, diacetoxyscirpenol, T2, HT2, and sterigmatocystin were analyzed using ultra-high-performance liquid chromatography–tandem mass spectrometry (UHPLC–MS/MS). Deoxynivalenol was tested using enzyme-linked immunosorbent assay (ELISA). Mycotoxins occurred in ranges and frequencies in the following order: sterigmatocystin (0–7 ppb; 74.5%), aflatoxin (0–993 ppb; 55.5%), citrinin (0–9 ppb; 55.5%), ochratoxin A (0–110 ppb; 30%), fumonisin (0–76 ppb; 26%), diacetoxyscirpenol (0–24 ppb; 20.5%), and combined HT2 + T2 (0–62 ppb; 14.5%), and deoxynivalenol was detected in only one sample at 510 ppb. Overall, low amounts of toxins were observed in rice with a low frequency of samples above the regulatory limits for aflatoxin, 13.5%; ochratoxin A, 6%; and HT2 + T2, 0.5%. The maximum co-contamination was for 3.5% samples with six toxins in different combinations. The rice cultivar, paddy environment, time of harvest, and millers influenced the occurrence of different mycotoxins. There is a need to establish integrated approaches for the mitigation of mycotoxin accumulation in the Kenyan rice.

## 1. Introduction

Rice (*Oryza sativa* L.) is an important food crop that contributes approximately 21% of world per capita caloric intake [1]. In Africa, it holds enormous potential as a food crop and suits within the African Union’s Comprehensive Africa Agriculture Development Programme (CAADP) strategy for achieving food and nutritional security across the continent [2]. In Kenya, rice is the third most important cereal after maize and wheat and is cultivated as a semi-subsistence crop mainly by small-scale farmers [3]. The current Kenyan annual rice production is estimated to be 110,000 tonnes (KNA, 8 January 2020: https://www.kenyanews.go.ke/ (accessed on 10 March 2021)) and only meets 20% of the country’s demand [4]. The demand is increasing at an annual rate of 12% due to changes in consumer preference, population growth, urbanization, and other changes in lifestyles, which stipulate the need for less fuel and rapid cooking methods [3]. The increased rice demand implies that there is a need to adopt practices that enhance the quantity and quality of the grain to enhance food security and safety in the country.

The most common loss of grain quality occurs due to colonization and contamination by molds [5]. Colonization of grains by some fungi can lead to kernel rot and contamination by toxic secondary metabolites called mycotoxins. The fungal contaminants of grains are either pathogenic or saprophytic species that infect and colonize cereal crops before, during, or after harvest [6]. For field pathogenic fungi, the ability to colonize the plant is dependent on the presence of virulence factors and the conditions that aggravate the susceptibility of the plant [7,8]. Good agronomic practices can reduce plant stress and hence lessen the susceptibility of the crop to opportunistic fungi, including those that produce mycotoxins [9]. For fungi that attack stored grains, the ability to colonize and produce toxins is mainly dependent on water activity and temperature [10]. Drying of rice grain to <14% moisture content limits fungal growth and colonization during storage [11].

The fungi that colonize and contaminate rice with mycotoxins and the potential risk of the contaminants to humans and other animals are shown in Table 1. Across crop value chains, there are variations in point(s) at which certain mycotoxins occur. Fumonisin has been reported in the field, and it is unclear whether it accumulates during grain storage [12]. Other mycotoxins, including aflatoxin, citrinin, deoxynivalenol, ochratoxin A, sterigmatocystin, T2, and HT2, have been detected in different crops before and after harvest [13,14,15,16,17]. Based on chemical structures, these fungal metabolites are grouped into sub-types, which may occur together or individually in a contaminated sample. Aflatoxins that contaminate cereals and oil crops are grouped into four types: B1, B2, G1, and G2. Aflatoxin B1 is the most potent naturally existing carcinogen known to humans [18]. Citrinin (also called monascidin A) is historically recognized by its yellow crystals named “yellow rain” and suspected in an Asian biowarfare [19]. The fumonisin B analogs, comprising toxicologically important fumonisin B1, fumonisin B2, and fumonisin B3, are naturally occurring in food and feed commodities, with fumonisin B1 being the most predominant [20]. Trichothecenes are grouped into four types, A, B, C, and D [21]. Trichothecenes of importance in food and feeds include diacetoxyscirpenol, T2, HT2 (type A), and deoxynivalenol (type B) [21]. Ochratoxins are grouped into A, B, and C [22]. Ochratoxin A is most prevalent in cereal grains, dried fruits, wine, and coffee [23]. Sterigmatocystin is a precursor in the formation; shares biological activity, including carcinogenic effects; but is less toxic than aflatoxin B1 [24]. Sterigmatocystin has been reported in many feed and foodstuffs [25,26].

The hot and humid tropical climate provides ideal conditions for the growth of most mycotoxigenic fungi, but most of the mycotoxin studies in sub-Saharan Africa (SSA) have focused on aflatoxin contamination in maize and peanuts [27,28,29,30]. Information on the colonization and/or contamination of rice by mycotoxin is limited in most rice-growing countries of SSA [31,32,33]. A recent survey showed that Kenyan rice grains were colonized by multiple mycotoxin-producing fungi [33]. Unfortunately, Kenya, like most other developing countries in SSA, lacks the capacity for systematic monitoring and does not have regulatory standards for most mycotoxins, except for aflatoxin at 10 ppb (10 µg/kg) [27]. Emerging food safety concerns due to other mycotoxins imply the need for more stringent approaches to safeguard rice consumers.

Owing to the increasing demand for rice in Kenya, there is a need to monitor its grain quality. It has previously been claimed that mycotoxin contamination in rice is usually lower compared with that in wheat or maize [34]. However, there is a dearth of information about the occurrence, levels of mycotoxins, and associated factors for Kenyan rice. Most (80%) of the rice is produced under irrigation, which is mainly managed by the Kenyan National Irrigation Authority (NIA) [35]. Recent studies have shown a high prevalence of mycotoxigenic fungi in rice and have raised concerns of a potential human exposure to mycotoxins, which could have adverse irreversible health problems [13]. This study was conducted as part of the NIA’s efforts to improve the rice value chains through the provision of evidence-based advice to local communities and to all stakeholders who collaborate in promoting the marketing, safe storage, and processing of agroproduce grown on all public and community-based irrigation schemes. The objectives of the study were to assess the extent of rice grain contamination and co-contamination with mycotoxins, and to identify the factors that are associated with the occurrence of individual mycotoxins in the Mwea Irrigation Scheme, a major rice-producing region in Kenya.

**Table 1 toxins-13-00203-t001:** Major mycotoxin-producing fungi, health impacts, and global regulatory limits.

Mycotoxin	Major Producing Fungi [12,14,16,23,36,37,38,39,40,41]	Human Health Impact [18,42,43,44,45,46,47,48]	Regulatory Limit (ppb) in Human Food [24,49,50,51,52,53,54]
Aflatoxin	*Aspergillus flavus*, *A. parasiticus*, and *A. nomius*	Liver cancer	10
Citrinin	Many species of *Aspergillus* (e.g., *A. niger*), *Penicillium* (e.g., *P. citrinum*), and *Monascus* (e.g., *M. pallens*)	Potential carcinogen and nephrotoxicity	100
Diacetoxyscirpenol	*Fusarium* spp. (mainly *F. langsethiae*, *F. poae*, *F. sporotrichioides*, and *F. sambucinum*)	Vomiting alimentary toxic aleukia	100
Deoxynivalenol	*Fusarium* spp. (e.g., *Gibberella zeae*)	Nausea and vomiting	1000
Fumonisin	*Fusarium* spp. (e.g., *F. verticillioides* and *F. proliferatum*)	Esophageal cancer	2000
HT2	*Fusarium* spp. (e.g., *F. langsethiae*, *F. poae*, and *F. sporotrichioides*)	Alimentary toxic aleukia	50
T2	*Fusarium* spp. (e.g., *F. langsethiae*, *F. poae*, and *F. sporotrichioides*)	Alimentary toxic aleukia	50
Ochratoxin A	*Aspergillus ochraceus*, *A. carbonarius*, *A. niger*, and *Penicillium verrucosum*	Potential human carcinogen and kidney damage	5
Sterigmatocystin	*Aspergillus versicolor*	Esophageal and lung cancer	NR

NR: No set maximum limit.

## 2. Results

### 2.1. Description of Datasets and Its Distribution

Mycotoxins of importance to human and livestock health were analyzed in rice samples (*n* = 200) that were collected in the major Kenyan rice-growing region of Mwea, Kirinyaga County (Table 1 and Figure 1). The collected samples were milled grains of two rice cultivars, namely, Pishori (hereinafter referred to as NIBAM 11, an aromatic cultivar that is popularly grown for sale to local markets in Kenya) and BW 196 (hereinafter referred to as NIBAM 109, a high-yielding and pest-tolerant nonaromatic cultivar that is mainly cultivated for subsistence use in Kenya). The rice samples were provided by random resident small-scale traders (who obtain the milled rice from the miller and sell it to on-transit customers in different quantities, ranging 2 to 50 kg) at different market centers in the Mwea Irrigation Scheme, Kenya. All mycotoxins, except for deoxynivalenol, were tested on all samples using LC–MS/MS protocols, which were optimized to detect multiple mycotoxins (Appendix A). In addition, deoxynivalenol was tested in the same rice samples using ELISA. For statistical analysis and reporting of the LC–MS/MS data, the amounts of subtypes of individual mycotoxins were summed into the specific toxin and designated as follows: fumonisin B1 + fumonisin B2 + fumonisin B3 = fumonisin; aflatoxin B1 + aflatoxin B2 + aflatoxin G1+ aflatoxin G2 = aflatoxin. HT2 and T2 were also combined into HT2 + T2 (Table 2). Because the levels of contamination by all mycotoxins were low, and the data were not normally distributed, the main summaries presented included the percentages of samples above the detection and regulatory limits, range, median, and the 75th quartile (Table 2). Because only one sample had detectable deoxynivalenol (with 510 ppb), the toxin was dropped from the testing of the association with other factors.

### 2.2. Mycotoxin Profiles in Rice Samples from the Mwea Irrigation Scheme

Nine mycotoxins were analyzed and detected at different frequencies (Table 2). The highest frequency (74.5%) was in sterigmatocystin with levels in the range of 0.03–7 ppb. There is no documented regulatory limit for sterigmatocystin. Second in frequency was citrinin, which was detected in the range of 0.1–9 ppb in slightly over half of the samples. Again, no samples had citrinin contamination above the EU regulatory limit of 100 ppb. Third was aflatoxin in slightly over half of the samples and in the range of 0.26–993 ppb. Most of the samples had very low aflatoxin contamination, and 86.5% of the samples were below the Kenyan legal limit of 10 ppb. Interestingly, contamination with aflatoxin G1 and G2 was below 1 ppb in all samples tested. A few samples had high levels of aflatoxin B1 contamination (maximum 921 ppb) and B2 (maximum 72 ppb). Owing to the low average recovery (52%) for aflatoxin B2, the values reported herein might be lower than the actual contamination (Table 2 and Appendix A). Diacetoxyscirpenol was detected in 21% of the samples in the range of 0.05–24 ppb, and none of the samples were above the regulatory limit of 100 ppb. Fumonisin was detected in 26% of the samples in the range of 2.35–76.0 ppb, and none was above the regulatory limit of 2000 ppb. HT2 + T2 was detected in the range of 0.04–62 ppb at a frequency of 14.5%, and only one sample was above the EU regulatory limit of 50 ppb. Ochratoxin A was detected in the range of 0.19–111 ppb at 30% frequency with 6% of the samples being above the regulatory limit of 5 ppb (Table 2).

Co-contaminations were observed in different combinations (Table 3 and Appendix A). The two-way combinations were most common with a frequency ranging between 0% (deoxynivalenol and HT2 + T2) and 48% (sterigmatocystin and citrinin) (Table 3). The frequency of co-contamination by both aflatoxin and sterigmatocystin was second to the highest at 46%, followed by citrinin and aflatoxin (35%), and then a quarter of the samples had a co-contamination of sterigmatocystin and ochratoxin A (Table 3). Co-contaminations with the rest of the pairs of toxins were observed in less than a quarter of the samples. Co-contamination by more than two toxins was observed in different combinations at varying frequencies as follows (Appendix A): first, 111 samples (55%) had at least three toxins; second, 66 samples (33%) had at least four toxins; and third, 27 samples (13.5%) had at least five toxins. The maximum co-contamination was for 11 samples (3.5%), which had six toxins in different combinations (Appendix A).

A nonparametric correlation test showed positive relationships between pairs of some mycotoxins (Table 3). A positive association was observed between sterigmatocystin and aflatoxin (*ρ* = 0.19, *p* = 0.0065) or citrinin (*ρ* = 0.26, *p* = 0.0002). Similar relationships were observed between ochratoxin A and aflatoxin (*ρ* = 0.15, *p* = 0.0377), citrinin (*ρ* = 0.17, *p* = 0.0149), and diacetoxyscirpenol (*ρ* = 0.15, *p* = 0.0294). T2 positively correlated with HT2 (*ρ* = 0.15, *p* = 0.0307). HT2 + T2 was also positively correlated with fumonisin (*ρ* = 0.23, *p* = 0.001), diacetoxyscirpenol (*ρ* = 0.21, *p* = 0.0028), and ochratoxin A (*ρ* = 0.22, *p* = 0.0019).

### 2.3. Mycotoxin Profiles in NIBAM 11 and NIBAM 109 Rice Cultivars

Samples of NIBAM 11 (*n* = 184) and NIBAM 109 (*n* = 14) cultivars were tested for the eight mycotoxins. The cultivars differed in frequency of contamination by aflatoxin *X*^2^ (2, *n* = 198) = 40.4, *p* < 0.0001; fumonisin *X*^2^ (1, *n* = 198) = 12.2, *p* = 0.0005; and HT2 + T2 *X*^2^ (1, *n* = 198) = 5.1, *p* = 0.0188 (Table 4). The frequency of aflatoxin was eight times more in NIBAM 11 than in NIBAM 109. Similarly, fumonisin occurrence was five times higher in NIBAM 11 than in NIBAM 109. There were no detectable HT2 + T2 levels in NIBAM 109, while the toxin was detected in 22% of NIBAM 11 samples (Figure 2).

### 2.4. Effects of Paddy Harvesting Date on the Occurrence of Mycotoxins

Paddy harvest date (or season) affected the likelihood of contamination of rice by fumonisin *X*^2^ (3, *n* = 198) =16.9, *p* = 0.0007, and sterigmatocystin *X*^2^ (3, *n* = 198) = 11.1, *p* = 0.0113 (Table 4). Rice harvested in October 2018 did not have detectable levels of fumonisin. However, rice harvested in subsequent months of 2018 had fumonisin, with three times the frequency of contamination in December compared with that in November 2018. The frequency of rice contamination with sterigmatocystin increased from 25% of the samples in October 2018 to 92% in November 2018 before dropping to 76% in December 2018 (Figure 3). Rice harvested in January 2019 did not have detectable fumonisin and sterigmatocystin.

### 2.5. Effects of Pre- and Postharvest Management and Handling on the Occurrence of Mycotoxins

Contamination of rice by mycotoxins was influenced by the pre- and postharvest handling environments and/or processes, including the paddy cultivation site and the miller (Table 4). The paddy environments had significant effects on the likelihood of rice contamination by diacetoxyscirpenol *X*^2^ (14, *n* = 123) = 24.6, *p* = 0.0387 (Table 4 and Figure 4). Within the paddy cultivation sites, diacetoxyscirpenol was not detected in six sites (which represented 18% of all the samples) (Figure 4). For the rest of the samples across the remaining nine cultivation sites, diacetoxyscirpenol occurred at frequencies ranging from 10% to 55% (Figure 4).

Aflatoxin occurrence varied significantly among the samples that originated from different millers *X*^2^ (48, *n* = 190) = 66.7, *p* = 0.0382 (Table 4 and Figure 5). Aflatoxin was not detected in three mills (which represented 4.7% of all samples). For the rest of the millers, the frequency of contamination ranged from 25–100%. Three millers had detectable aflatoxin in all samples. It should be noted that the sample size for some of the millers was small (mainly dependent on the willingness of the traders to participate), and this could have affected the likelihood of detecting contamination.

## 3. Discussion

This study focused on identifying the mycotoxin profiles in white rice collected from the Mwea Irrigation Scheme of Kenya, where 80% of the rice that is consumed in Kenya is cultivated. The design of the study enabled us to get a snapshot of the level and frequency of contamination during different harvesting stages of rice. For diagnostics, an LC–MS/MS protocol was optimized for the detection of multiple mycotoxins in milled rice. The protocol was used to identify co-contamination of rice with low levels of up to six different mycotoxins in some of the samples. To the best of our knowledge, this is the first study to report multiple mycotoxins in rice produced in Kenya. These findings are indications of chronic consumer exposure to low levels of different mycotoxins in rice. This study provides evidence of potential food safety concerns and calls for an urgent establishment of a policy on safe rice production through better practices along the value chain.

Low levels of contamination were observed for all mycotoxins, except for aflatoxin and ochratoxin A, which had a low percentage of samples above the limits allowed in human food (or regulatory limits). These low levels and frequency agree with a previous report that rice has less mycotoxin burden than maize [55]. It should be noted that Kenya does not have any set regulatory limits for all mycotoxins except aflatoxin at 10 ppb [54]. The ochratoxin A regulatory limit set by the European Union is 5 ppb [50]. Previous surveys mainly focused on mycotoxins in other crops, including maize, where there were frequent reports of high levels of contamination by aflatoxin and/or fumonisin [28,30,56]. High levels of aflatoxin contamination have also been reported in peanuts [29]. A recent survey reported a high frequency of aflatoxin-producing fungi in rice samples from local markets in Kenya and implied that consumers were at risk of exposure to these damaging toxins [33]. The findings in the current study indicate that the levels and frequencies of contamination of rice by aflatoxin are reasonably lower than those reported in maize and peanuts in Kenya. These findings should not be interpreted as suggesting that the consumption of rice with lower levels of aflatoxin contamination is safe; rather, there is likely a health risk associated with the long-term consumption of such rice. A long-term exposure to small amounts of carcinogenic substances (as in the case of these mycotoxins) could cause different types of cancer [23,44,57,58].

Co-contamination of rice with multiple mycotoxins was observed, with the highest being for six toxins in 3.5% of the samples. Co-contaminations could occur because of coinfection of rice with different mycotoxigenic fungal species and/or in vivo biotransformation of related toxins [33,59]. For example, colonization of rice by multiple fungal species, including producers of aflatoxin (*A. flavus*, *A. parasiticus*, and *A. nomius)* and sterigmatocystin (*A. versicolor*), was recently reported [31,32,33]. If co-colonization occurs under conditions that favor the production of specific toxins by the fungi, then co-contamination could be observed. On the other hand, biotransformation of mycotoxins has previously been reported (e.g., an in planta conversion of T2 to HT2) [59]. In the current study, a high co-contamination occurred between sterigmatocystin and aflatoxin or citrinin. The observed co-occurrence and the positive correlation between sterigmatocystin and aflatoxin are expected because of shared biosynthetic pathways and water activity of the producing fungi [60,61]. Based on recognized growth characteristics of different fungi in response to water activity, co-contamination of rice with mycotoxins that are produced by fungi with distinct water activity requirements (hygrophilic vs. xerophilic) could only occur if rice grains were infected by different fungal species at alternate stages of production and handling [62,63]. Indeed, the positive correlations among toxins that are produced by xerophilic fungi (mainly *Aspergillus* species, including aflatoxin, ochratoxin A, and sterigmatocystin) and hygrophilic fungi (mainly *Fusarium* species; HT2 + T2 and diacetoxyscirpenol) are suggestive of possible effects of water activity. Animal studies have shown that there is a higher risk of liver cancer with co-exposure to aflatoxin and fumonisin [64].

The frequency of contamination with aflatoxin, fumonisin, and HT2 + T2 was significantly and consistently much lower in NIBAM 109 than in NIBAM 11 cultivar. Here, it should be noted that aflatoxin is produced by xerophilic *Aspergillus* species, while the rest are produced by hygrophilic *Fusarium* species [62,63]. Thus, based on shared fungal genera and similarity of water activity, the trend of cultivar response to fumonisin and HT2 + T2 was expected, but not to aflatoxin. This similarity in response of the two cultivars shows that NIBAM 109 is resistant to colonization and contamination by the fungal species that produce the three mycotoxins. Similarities in fungal effector repertoire have been reported and could explain the relatedness in plant defense to infection by fungal pathogens [65]. Although NIBAM 109 was found to be less vulnerable, NIBAM 11 is more popular among consumers [66]. However, NIBAM 11 has been reported to be susceptible to many biotic stresses, including blast disease, rodents, and stem borers, and to abiotic stresses, such as excess nitrogen in the soil, which leads to lodging [3]. A strong correlation has been reported between susceptibility of maize to damage by insects and contamination of crop plants by mycotoxins [67]. Although we did not correlate lodging with mycotoxin contamination, we presume that this inherent lodging trait of NIBAM 11 could cause the paddy to come into contact with soil, hence increasing the risk of colonization by mycotoxigenic species. It should be noted that our sampling for NIBAM 109 was affected by the existence of very few samples in the market. A further robust sampling and controlled experiments are needed to enhance comparisons of the resistance to mycotoxin accumulation between the two rice cultivars.

Differences in contamination of rice by diacetoxyscirpenol in paddy environments were observed. Like other *Fusarium* mycotoxins, diacetoxyscirpenol is mainly produced in the field and could increase during storage of cereals [68]. Previous studies have shown that the production of trichothecenes is modulated by environmental factors, such as soil nutrients, ambient temperature, water activity, and fungal genetics [68]. It has also been observed that specific farming methods, such as rotation with specific crop species (e.g., soybean) could reduce the occurrence of *Fusarium* and trichothecenes in wheat [69]. It has also been argued that kernel chemical composition, which is influenced by soil nutritional content, can affect the susceptibility of cereals to mycotoxins [70]. Besides soil and climatic conditions, multiple localized or region-based preharvest agronomic practices could influence the occurrence of mycotoxigenic fungi. Except for the differences in sampling dates reported herein, data on the agronomic practices in paddy production environments at different stages of rice growth were not collected. A sampling bottleneck might have contributed to the observed differences in fumonisin and sterigmatocystin. There is a need to conduct further longitudinal surveys and field experiments to determine the preharvest and storage practices, which could be fueling the occurrence of mycotoxins in Kenyan rice.

Significant differences in the frequency of aflatoxin were observed in rice from different millers. This observation could imply a lack of appropriate grain handling or a purchase of low-quality rice from farmers who do not practice proper postharvest handling. We did not investigate the posthandling practices of the traders and the millers in this study. Based on our experiences with mycotoxins in other crops, the potential avenues for aflatoxin contamination could include dropping of the paddy on the soil after harvest, heaping of damp paddy during transportation or storage, inadequate drying (storage at >14%), kernel breakage during milling, and storage in sheds with inadequate ventilation. Differences in milling and storability properties have been reported for different cultivars of rice [71]. In Kenya, approximately 10% of the grain is reported to be broken during milling [3]. Kernel breakage caused by poor postharvest handling of kernel and grains could increase infection and exacerbate fungal contamination [72].

Sustainable food security initiatives must consider the safety of consumers through reliable surveillance and monitoring of the quality of produced and consumed commodities. Here, we have provided evidence of the existence of multiple mycotoxins in a Kenyan staple food, and we anticipate that this report can be a harbinger to establish policies and standards to ensure the safety of rice consumers. The countries in sub-Saharan Africa must endeavor to establish policies that can guide regular empirical food quality assessments, as this would enhance and guarantee a healthy population with better standards of living and sustained development for prosperity. We propose the adoption of integrated strategies across different points of the value chain to avoid human exposure to mycotoxins in consumed rice. At the preharvest stage, good agronomic practices could be adopted to prevent the entry of toxigenic fungi by adopting optimal practices to ensure a healthy rice crop [9]. At the postharvest stage, contamination could be reduced by the avoidance of heaping of damp paddy during transportation or storage, adequate drying of the grain up to <14% before storage, avoidance of kernel breakage during milling, and storage of milled rice in hermetic bags and under ventilated sheds [73]. Additionally, millers should be encouraged to create awareness for the farmers to adequately dry the grain to below 14% moisture content before delivery for subsequent processing at the mill. Finally, consumers could be advised to adopt dietary diversification to reduce intake of foods derived from most vulnerable crops.

## 4. Materials and Methods

### 4.1. Study Site

This survey was conducted in the Mwea Irrigation Scheme, Kirinyaga County, Kenya. The scheme is located approximately 105 km north of Nairobi and was established in 1956. The region is approximately 1190 m above sea level. It lies within latitude 37°13′ E and 37°30′ E and longitude 0°32′ S and 0°46′ S. There are two rainy seasons: a short rainy season (October to November) and a long rainy season (April to May). The mean annual rainfall is about 930 mm [35], and temperature ranges between 14 °C and 31 °C, while the relative humidity ranges from 55% to 70% [35]. The scheme is characterized by basin irrigation systems, which consist of earthen canals feeding into basins. The main paddy crop is cultivated between June and December. After harvest, most farmers manage a ratoon crop for over two months (end of February to March) every year. The Mwea Irrigation Scheme produces about 90,000 tonnes of paddy rice on 26,000 acres annually and is the largest (80%) producer of rice in Kenya [74].

### 4.2. Sampling of Rice

We conducted a cross-sectional survey of the occurrence of multiple mycotoxins in marketed milled white rice within five rice marketing suburbs (Wang’uru, Kagio, Kandongu, Kimbimbi, and Mutithi) of the Mwea Irrigation Scheme, Kirinyaga County, Kenya, between February and May 2019. We trained and recruited research assistants who collected samples of milled rice from random small-scale traders (*n* = 200; these belonged to a group that obtains the milled rice from the miller and sells it to on-transit customers in different quantities, ranging from 2 to 50 kg) within a radius of 30 m from each of 25 milling points, which were strategically distributed within the five suburbs. The collected samples were milled grains of two rice cultivars, namely, Pishori (hereinafter referred to as NIBAM 11, an aromatic cultivar that is popularly grown for sale in Kenyan local markets) and BW 196 (hereinafter referred to as NIBAM 109, which has a higher grain yield and is more tolerant to pests than Pishori but is nonaromatic and hence is not preferred by most Kenyan consumers; farmers grow it mainly for subsistence use). The traders kindly provided 250 g from a 50 kg bag and related information on the rice cultivar, season of harvesting of the paddy, and locality where the paddy was grown. Because of similar handling practices during and after the harvest of paddy within the region, data about these practices were not collected. Samples were kept in a freezer at 4 °C until the time of mycotoxin testing at the Mycotoxin and Nutritional Analysis Platform of the International Livestock Research Institute (ILRI), Nairobi, Kenya.

### 4.3. Milling of Rice Grain in the Laboratory

All rice samples were ground and homogenized into a fine flour of approximately 0.5 mm diameter granules and subsampled into three subsets using the Romer Series II Mill (Romer Labs Diagnostic GmBH, Tulln, Austria). From these subsamples, a 5 g flour was weighed into a 50 mL Falcon tube and used in the rest of the assays.

### 4.4. Mycotoxin Analysis

Mycotoxins were analyzed using enzyme-linked immunosorbent assay (ELISA) and liquid chromatography with tandem mass spectrometry (LC–MS/MS). Deoxynivalenol was analyzed using ELISA.

#### 4.4.1. Mycotoxin Analysis Using ELISA

Deoxynivalenol was extracted from a 5 g subsample of rice flour using sterile distilled water and analyzed in two replicates using a commercially available ELISA kit following the manufacturer’s protocol (Helica Biosystems, Inc., Santa Ana, CA, USA). The solid-phase direct competitive ELISA kit (Helica catalogue number 941DON01M-96) consisted of a polystyrene 96-well microplate coated with a monoclonal antibody that was optimized to bind to deoxynivalenol, resulting in varying yellow color intensities based on the amount of the toxin. The lower and upper limits of detection of the kit were 0.5 and 10 ppm, respectively, based on information provided by the supplier.

#### 4.4.2. Mycotoxin Analysis of Samples Using LC–MS/MS

##### Chemicals and Reagents

Chromasolv-grade methanol (MeOH), acetonitrile (ACN), ammonium acetate formic acid, and quantification standards, including an aflatoxin mix solution (GI, G2, B1, and B2), a fumonisin mix solution (B1 and B2), a trichothecenes mix solution (diacetoxyscirpenol, HT2, and T2), ochratoxin A, fumonisin B3, citrinin, and sterigmatocystin, were used. The mycotoxin quantification standards were obtained from Sigma Aldrich (St. Louis, MO, USA) through a Kenyan chemical supplies dealer.

##### Preparation of Mycotoxin Standards for Use in LC–MS/MS

From each standard, 100 µL was prepared in 1900 µL of acetonitrile. A standard working solution was prepared by mixing single standards of mycotoxin with acetonitrile. The working solution was then used in the preparation of calibration standards. The standards and the working solution were stored at −20 °C.

##### Sample Preparation and Extraction

For LC–MS/MS analysis, mycotoxins were extracted using a 20 mL extraction solution (acetonitrile/water/formic acid, 79:20:1, v/v/v) on a shaker for 90 min at 220 rpm. After extraction, 700 µL of the extract was transferred to Eppendorf tubes containing 700 µL of mobile phase A (formic acid, ammonium acetate, and water). The solution was mixed thoroughly by vortexing and transferred into HPLC vials after microfiltration using 0.20 µm nylon microfilters. The content of the vial (10 µL) was injected directly into the LC–MS/MS system.

##### Sample Analysis in LC–MS/MS

Identification and quantification of compounds was performed using the Shimadzu Nexera liquid chromatograph system coupled to the LC–MS/MS 8050 triple quadrupole mass spectrometer detector (Shimadzu Corporation, Kyoto, Japan). The whole system consisted of the SIL-30AC autosampler, LC-20AD solvent delivery pump, column oven, and 8050 triple quadrupole detector. Separation was performed at 40 °C using a Synergi Hydro-RP analytical column (2.5 µm particle size, 100 mm × 3 mm) (Phenomenex, Torrance, CA, USA) operating at a flow rate of 0.4 mL/min. A binary mobile phase consisting of phase A (water/formic acid, 99:1, v/v) and 10 mM ammonium acetate and phase B (methanol/water/formic acid, 97:2:1, v/v/v) and 10 mM ammonium acetate was used. The gradient elution program was set as follows: 0 min 5% B, 4.5 min 50% B, 8 min 100% B, 10 min 100% B, and 12 min 5% B, held for a further 2 min for re-equilibration, giving a total run time of 14 min. The detection by MS/MS was performed on the 8050 triple quadrupole mass spectrometer equipped with an electrospray ionization source operating in both positive and negative ionization modes. The ionization source was operated under the following conditions: nebulizing gas and drying gas flow rates of 3 L/min and 10 L/min, respectively; interface voltage of 4.5 kV; desolvation line temperature of 300 °C; and heating block temperature of 400 °C. Direct infusion of individual neat standards into the mass spectrometer was performed to identify the toxins’ transitions in multiple reaction monitoring (MRM) mode for quantitation. Optimized MRM parameters and the corresponding toxin retention time are shown in Appendix A. Analytical data were called using LabSolutions software version 5.89 (Shimadzu Corporation, Kyoto, Japan, 2014).

### 4.5. Data Analysis

Descriptive statistics was conducted to determine the percentage of samples with different levels of contamination (based on the method’s detection and the existing regulatory limits) by each of the toxins. Publicly available data on regulatory limits for individual toxins were used to categorize samples as above or below the regulatory limits. Because Kenya or East Africa does not have set limits for mycotoxins, except for aflatoxin, European standards were used to assign the samples into different categories (Table 1). A binary coding was established to enhance testing for associations with other survey factors. Mycotoxin levels were coded as follows: samples with contamination equal to or above the limit of detection were assigned “1” to mean toxin is present, while those below the limit were assigned “0” to mean clean or undetectable. Because of the marginal number of samples with contamination above the regulatory limit of each mycotoxin, no statistical model was implemented for the drivers of contamination above the limit. A binomial logistic regression was used to determine which of the factors (date of harvest of paddy or “season,” millers, locality of millers, cultivar, and site where the cultivar was grown) were associated with the presence of each toxin. All factors were entered into and removed sequentially from the model until the best combinations, based on low Alkaike information content and high coefficient of determination, were retained. A summary of tables or figures was generated for factors that showed significant differences in likelihood of contamination by each of the toxins. All statistical analyses were implemented in JMP PRO version 15.2 (SAS Institute Inc., Cary, NC, USA, 1989–2019).

## Figures and Tables

**Figure 1 toxins-13-00203-f001:**
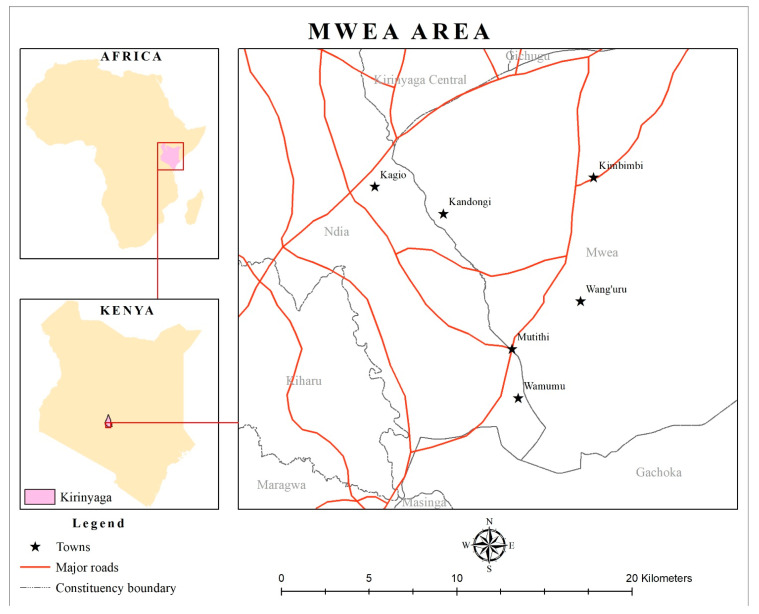
A map of Kenya showing administrative counties. Zoomed in is Kirinyaga County, where sampling was conducted for the analysis of multiple mycotoxins.

**Figure 2 toxins-13-00203-f002:**
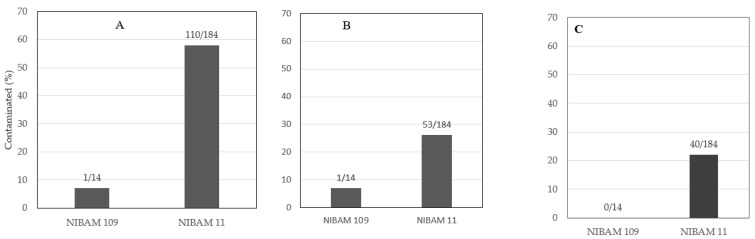
Mycotoxin contamination in two cultivars of rice: (**A**) aflatoxin, (**B**) fumonisin, (**C**) HT2 + T2. On top of the bars are the numbers of contaminated (numerator) out of the total numbers (denominator) for the specific cultivar shown in the *x*-axis.

**Figure 3 toxins-13-00203-f003:**
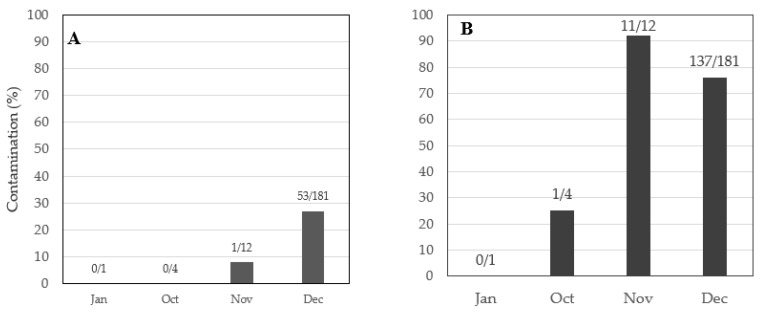
Effect of harvesting date (seasons) on mycotoxin contamination in rice: (**A**) fumonisin, (**B**) sterigmatocystin. The numerators on top of the bars represent numbers of samples with contamination out of those harvested (denominator) in the months of 2018 (October, November, and December) and 2019 (January) shown on the *x*-axis.

**Figure 4 toxins-13-00203-f004:**
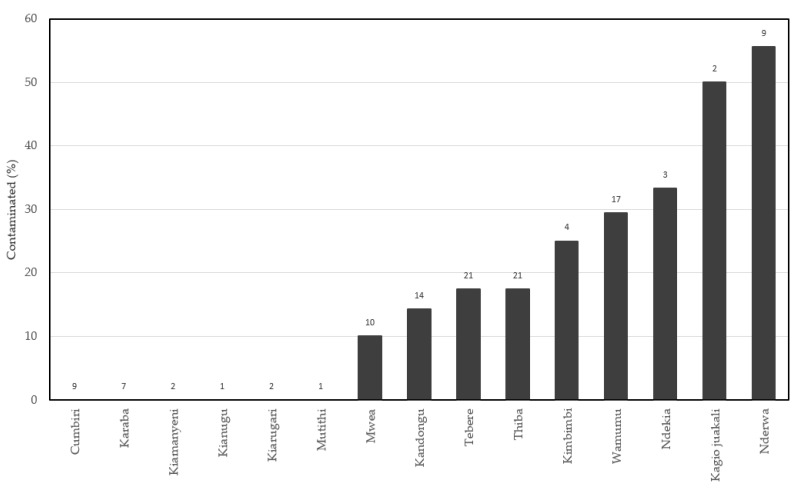
Prevalence of diacetoxyscirpenol in paddy environments within the Mwea Irrigation Scheme of Kirinyaga County, Kenya. The number on top of the bar is the sample size from each of the sites shown in the *x*-axis.

**Figure 5 toxins-13-00203-f005:**
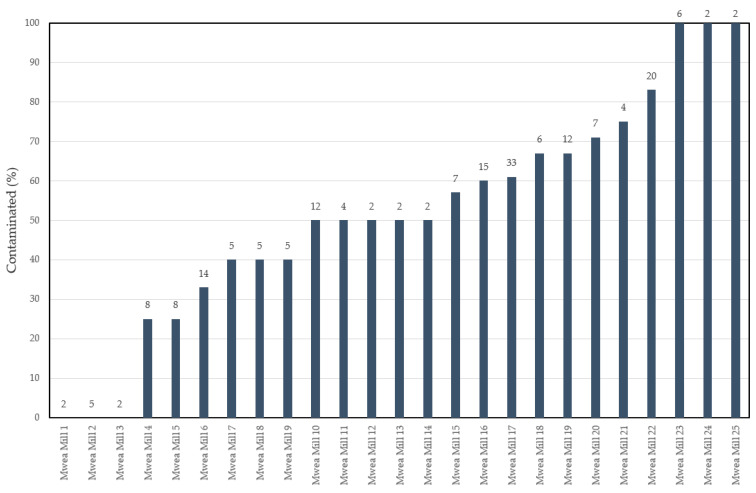
Frequency of aflatoxin in rice grain samples from different millers within the Mwea Irrigation Scheme of Kirinyaga County, Kenya. The number on top of each bar is the sample size associated with each of the millers shown in the *x*-axis. Rice millers are coded as follows: “Mwea Mill followed by a serial number.”

**Table 2 toxins-13-00203-t002:** Occurrence of mycotoxins in rice samples collected from the Mwea Irrigation Scheme.

Toxin	Sample Contamination (ppb)					
	LOD	Samples with Detectable Toxin (*n*; %)	Range	Median	75th Percentile	>ML (*n*; %)
Aflatoxin (AF) ^†^	0.26	110; 55.5	0–993	0.2	1.7	27; 13.5
Aflatoxin G2	0.1	3; 1.5	0–0.6	0	0	NA
Aflatoxin G1	0.05	6; 3	0–1	0	0	NA
Aflatoxin B2	0.07	22; 11	0–72	0	0	NA
Aflatoxin B1	0.04	108; 54	0–921	0.2	1.7	NA
Citrinin	0.1	111; 55.5	0–9	0.2	0.4	0
Diacetoxyscirpenol	0.05	41; 20.5	0–24	0	0	0
Deoxynivalenol ^¥^	500	1; 0.5	0–510	81	207	0
Fumonisin (FB) ^†^	2.35	52; 26	0–76	0	7.3	0
Fumonisin B1	1.53	0	0	0	0	NA
Fumonisin B2	0.15	1; 0.5	0–8	0	0	NA
Fumonisin B3	0.67	52; 26	0–76	0	6.3	NA
HT2 + T2 ^†^	0.04	29; 14.5	0–62	0	0	1; 0.5
HT2	0.03	6; 3	0–49	0	0	NA
T2	0.01	37; 18.5	0– 45	0	0	NA
Ochratoxin A	0.19	60; 30	0–111	0	0.5	12; 6
Sterigmatocystin	0.03	149; 74.5	0–7	0.2	0.3	NA

LOD = limit of detection. ^¥^ Analyzed using enzyme-linked immunosorbent assay, Helica Biosystems Inc. ^†^ Regulation is applied to the sum of the levels of individual toxins; hence, the percentage of samples above the maximum limit (ML) is indicated as not applicable (NA).

**Table 3 toxins-13-00203-t003:** Nonparametric correlations and co-occurrence of at least two mycotoxins in rice. Upper deck: percentage of co-occurrence. Lower deck: correlation test for mycotoxins (Spearman’s rho; *p*-value). Analysis was based on binary coding for presence, “1” or absence, “0” of each mycotoxin in 200 rice samples from the Mwea Irrigation Scheme, Kenya.

	Sterigmatocystin	Citrinin	Aflatoxin	Ochratoxin A	Fumonisin	Diacetoxyscirpenol	HT2 + T2	Deoxynivalenol
Sterigmatocystin		48	46	25	20	15	16	0.5
Citrinin	0.26; 0.0002		35	15	16.5	13.5	13	0.5
Aflatoxin	0.19; 0.0065	NS		20	15	12.5	11.5	0.5
Ochratoxin A	NS	0.17; 0.0149	0.15; 0.0377		6.5	9	10	0.5
Fumonisin	NS	NS	NS	NS		7	9.5	0
Diacetoxyscirpenol	NS	NS	NS	0.15; 0.0294	NS		7.5	0.5
HT2 + T2	NS	NS	NS	0.22; 0.0019	0.23; 0.001	0.21; 0.0028		0
Deoxynivalenol	NS	NS	NS	NS	NS	NS	NS	

NS: not significant.

**Table 4 toxins-13-00203-t004:** Factors affecting the occurrence of mycotoxins in rice from the Mwea Irrigation Scheme, Kenya. Likelihood ratio tests were implemented in a binomial logistic model with samples with detectable toxin level coded as 1 and those with undetectable coded as 0.

Model			Factor (*p*-Value)				
Toxin	*r* ^2^	*p*-Value	Paddy Harvest Date	Locality of Miller	Source of Paddy	Miller	Cultivar
Aflatoxin	0.4	0.0255	0.993	NI	0.4106	0.0382	<0.0001
Citrinin	0.16	0.4961	0.1575	0.6212	0.3398	0.4875	NI
Diacetoxyscirpenol	0.37	0.2341	0.3344	NI	0.0387	0.5012	0.1873
Deoxynivalenol	NA	NA	NA	NA	NA	NA	NA
Fumonisin	0.26	0.0369	0.0007	0.8425	0.3508	0.1313	0.0005
Ochratoxin A	0.21	0.1956	0.6977	0.0622	0.2952	0.1541	0.4661
Sterigmatocystin	0.33	0.6138	0.077	NI	0.4774	0.983	NI
HT2 + T2	0.16	0.3433	NI	NI	0.5955	NI	0.0188

NA = not applicable, as this was not included in the model because only one sample had a detectable amount of this toxin. NI = not statistically important and was dropped because it did not improve the model when included.

## Data Availability

Data is available from the corresponding author upon request. We choose not to make it public because of the privacy of the millers, whose business names are mentioned.

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
