# Peer review of "Multiple Mycotoxins in Kenyan Rice"

_toxins, 2021, doi:10.3390/toxins13030203_

Round 1
Reviewer 1 Report
1. I made a few comments for the manuscript. Please see the attached scanned PDF. I only scanned the pages that had corrections on them.
2. Table 2-The data for FB-ELISA does not make sense to me. If there was only FB detected in one sample, and the LOD is 50 ppb, how can the median be 4 ppb? I wonder if the FB-ELISA data should be included in this manuscript since it seems to have missed the many samples containing fumonisin B3, some fraction of which should have been detected by FB-ELISA. In addition, there are several mycotoxins showing a 0 median and 0 75th percentile. How is this possible?
3. The sample number for cultivar BW196 was much lower than Basmati 370. Are the statistical measurements reliable? Is it worthwhile to include the data from cultivar BW196?
4. Table 4. Is it possible to superscript the number 2 under model? In the legend describe the statistical model tested. I assume the model is the likelihood for the presence of the specific mycotoxin in the sample; is the model based on a minimum concentration? The legend should indicate what NA and NI mean.
5. Figure 3B. You might consider enlarging the Y-axis so that the 11/12 is not touching the top line of the graph.
6. Supplemental material. I applaud the authors for providing this material. I do wonder why the spike recovery of Aflatoxin B2 was 50%. Perhaps the authors should consider stating in the results section that the Aflatoxin B2 concentration in the samples might be larger than reported. Also please indicate why there is no data for spike recovery of the Sterigmatocystin. This is confusing since Sterigmatocystin was detected most frequently among the rice samples.
7. I do not see anything about biological replicates of each sample (perhaps I missed it). Standard error or standard deviation should be incorporated into the median values in table 2.

Author Response
We sincerely appreciate your comments and we have made efforts to address the issues raised. We believe your input has enabled us to improve on the quality of this manuscript. Our responses are shown below.
- I made a few comments for the manuscript. Please see the attached scanned PDF. I only scanned the pages that had corrections on them. Response: Thank you very much for these comments. We have made the proposed changes in the main text.
2. Table 2-The data for FB-ELISA does not make sense to me. If there was only FB detected in one sample, and the LOD is 50 ppb, how can the median be 4 ppb? I wonder if the FB-ELISA data should be included in this manuscript since it seems to have missed the many samples containing fumonisin B3, some fraction of which should have been detected by FB-ELISA. Response: we acknowledge the concerns raised about FB-ELISA. Based on the information from the manufacturer of the ELISA kit, the cross-reactivities of the fumonisins are as follows: B1 – 100%; B2 – 110% and B3 – 83%. While a cross-reactivity of 83% for FB3 is not very low in ELISA, it looks like the antibody did not perform well in detecting this toxin (as this was the most prevalent, based on LC-MS/MS data) in the current assay. Thus, we agree with the reviewer that FB-ELISA data can be omitted without causing a shift in the findings of this study. We have implemented the proposed change by removing the said data.
In addition, there are several mycotoxins showing a 0 median and 0 75th percentile. How is this possible? It is possible to have 0 median and 0 75th percentile if the data has many zeros, like in our case for some mycotoxins. The reason for this column was to reveal the skewness of the data. The range column shows the minimum and the maximum in each case.
3. The sample number for cultivar BW196 was much lower than Basmati 370. Are the statistical measurements reliable? Is it worthwhile to include the data from cultivar BW196?
Response: We would like to keep the finding of BW196 as this crop is important to farmers, but it is not as popular as Pishori (Basmati 370). The samples collected in this study are likely to be a reflection of the popularity of the cultivars among the traders, because very few farmers take BW196 to the market. BW196 is mainly cultivated for consumption by the farmers' households, and not for market as most buyers prefer the aromatic Pishori. By growing BW196, farmers fetch a a higher grain yield and the crop does not get as much damage from pests as those growing the aroma of Pishori, which attracts birds and rodents. We have edited the first paragraph of the Results and section 4.2 of Materials and Methods to capture this information.
For reliability of the statistical tests, we have included a statement (in the 4th paragraph of Discussion) on the caveat in the sampling of BW196, and a need a follow-up study to enhance better comparisons of the performance of these cultivars.
4. Table 4. Is it possible to superscript the number 2 under model? This has been modified to superscript.
In the legend describe the statistical model tested. I assume the model is the likelihood for the presence of the specific mycotoxin in the sample; is the model based on a minimum concentration? We have modified the legend as follows:
Table 4. Factors affecting the occurrence of mycotoxins in rice from Mwea Irrigation Scheme, Kenya. Likelihood ratio tests were implemented in a binomial logistic model with samples with and without detectable toxin level coded as 1 and 0, respectively.
The legend should indicate what NA and NI mean
We have provided the meaning of these abbreviations as footnotes.
5. Figure 3B. You might consider enlarging the Y-axis so that the 11/12 is not touching the top line of the graph.
Response: we have enlarged the figure.
6. Supplemental material. I applaud the authors for providing this material. I do wonder why the spike recovery of Aflatoxin B2 was 50%. Perhaps the authors should consider stating in the results section that the Aflatoxin B2 concentration in the samples might be larger than reported. Response: we have included a statement in the results section to show that aflatoxin B2 might have been higher than reported.
Also please indicate why there is no data for spike recovery of the Sterigmatocystin. This is confusing since Sterigmatocystin was detected most frequently among the rice samples. Response: thanks for noticing this mistake. We have now provided the spike-recovery data for fumonisin B1 and sterigmatocystin. This was an error of omission.
7. I do not see anything about biological replicates of each sample (perhaps I missed it).
Sampling issue: We only collected about 250 grams from each participant (a trader) and milled using the Romer Mill to homogenize and subsample into three sets. Our reporting is based on mycotoxins extracted from 5-gram flour of each of these subsamples. To enhance quality check in ELISA for deoxynivalenol, two extracts of each subsample were tested in the same ELISA plate. Similarly, we tested duplicates of each sub-sample after analysis of every 25 samples in LC-MS/MS. Given the method of sampling, I think these subsamples may not be referred to as biological replicates.
Standard error or standard deviation should be incorporated into the median values in table 2
These datasets are highly skewed and the best way to show the distribution is by measures of dispersion, diagnostic and regulatory limit bins (as shown in Table 2). We have attempted to explain this in the first paragraph of the results section.
Reviewer 2 Report
please see the attachment

Author Response
We sincerely appreciate your comments and we have made efforts to address the issues raised. We believe your input has enabled us to improve on the quality of this manuscript. Our responses are shown below.
Major issues:
1. Introduction is too long, and was written as a literature review. Suggest shorten it to under 2 pages by deleting paragraphs or literatures irrelevant to the objective of the study. For example, line 55-60, line 66-7
Response and action: we have edited paragraphs 2,3&5. The introduction section has now been reduced to 2 pages.
2. Figure 1 and Table 1 are unnecessary and should be deleted.
Response and action:
a) We would prefer to keep Figure 1 because the manuscript shall be read by people from other parts of the world, who might not have an idea of where Kenya or Africa is located on the global map.
b) Table 1 provides a summary for mycotoxigenic fungi, toxins they produce and the regulatory limits. This information would have taken a paragraph in this manuscript. Given that our study involved multiple mycotoxins, there was a need for us to provide this kind of review in a tabular format. Therefore, we will retain this table.
3. Materials and methods: The amount of sample (5g) for mycotoxin extraction is not representative. The recommended grain sample size for mycotoxin analysis is 25g in all official methods.
Response: We acknowledge that there is a major problem with sampling for mycotoxin analysis, and attempts have been made to establish standard analytical ways. Generally, larger samples would give more reliable measurements for mycotoxins (Whitaker et al. 2010). However, we observe that there are challenges in analysis of large samples due to constraints of resources. Given these challenges, we attempted to collect representative rice samples from different points of the 50-kg bag that the traders kept their commodity. From the pooled grain, a representative 250 gram mass was collected for analysis. We milled the whole 250 gram using a Romer Mill (a recommended sampling/sub-sampling and homogenizing device, which gives up to three milled sub-samples through different spouts). We tested mycotoxins in 5 grams of the flour of these sub-samples in a procedure that we have optimized and applied for analysis of aflatoxin for the last decade (see Mutiga et al. 2015 DOI: 10.1094/PHYTO-10-14-0269-R; Kizito et al. 2016; Dooso et al. 2019 https://doi.org/10.3390/toxins11080467). We have previously used the same protocol to report very high levels and frequency of aflatoxin in maize from Eastern Kenya (see Mutiga et al. 2014 DOI: 10.1094/PHYTO-01-14-0006-R). We believe these findings are informative and can serve the purpose of enlightening the communities about the safety of rice.
Minor issues:
4. Abstract may be too long. Thanks for highlighting this mistake. We have edited to conform to the journal requirements.
5. Line 136-145: This part should be in the Materials and Methods.
There is a always a need to provide some brief on the nature of samples in the first paragraph of the results section, particularly if this appears before the Materials and Methods section. We included this information because we thought it would make the results more understandable to the readers. We have edited this section and made sure that the wording is not a duplication of what is written in the materials and methods section.
6. Line 222: delete repeated p=. Thanks for spotting this typo. We have rectified it.
7. Line 242-244: The sentence is lack of clarity. Do you mean the sample size (number of samples) is too small to detect contamination? Response: we mean that the sample size was too small and we might have missed some contaminated samples. More samples would have given a more statistical power. We have edited to make this paragraph clearer.
8. This reviewer has difficulty to understand Table 3
We have rephrased the legend to read as follows:
Table 3. Non-parametric correlations and co-occurrence of at least two mycotoxins in rice. Upper deck: percentage of co-occurrence. Lower deck: Correlation test for mycotoxins (Spearman’s Rho; P-value). Analysis was based on binary coding for presence, “1” or absence, “0” of each mycotoxin in two hundred rice samples from Mwea Irrigation Scheme, Kenya.
9. Explanation of green and red dots are needed for Supplementary Figure 1.
We have provided the decode for the colors in the legend of this supplemental figure.
10. References: too many (90). Total number of references for a research article should not be more than 40. Unless we missed it in the Authors Instructions, there is no fixed number of references for a research article in Toxins. However, we have attempted to reduce the number of references while ensuring that all relevant literature (for all the mycotoxins tested) is provided in the manuscript.
Reviewer 3 Report
The Authors performed the research concerning detection of multiple mycotoxins in Kenyan rice samples. Food safety is a major public health issue worldwide. The climate on Earth is constantly changing, and in some parts of the globe these changes create favourable conditions for fungal growth and therefore mycotoxins production. The threat of these secondary metabolites occurance and contamination of food causes numerous health problems for humans as well as livestock, and is connected with a number of negative effects on the economy, such as crop losses and decrease in food quality. Therefore awareness, constant contamination monitoring and prevention is strictly necessary. I find this article a valuable position as it is a set of interesting data presenting, as the Authors stated, first report of the multiple mycotoxins co-contamination in rice in sub-Saharian Africa. The manuscript topic and content fits for publication in Toxins.
The idea of detection of multiple mycotoxins co-contamination in Kenyan rice is very important in order to create the awareness to ensure health security throughout the food chain and the results are valuable. The whole structure of the manuscript is reasonable and well prepared. The introduction briefly discusses the subject and introduces to the research, however in my in opinion the characteristics of mycotoxins could be done more detailed for all of them (Paragraph 5 in Introduction section). Experiments are correctly performed and results properly described. The manuscript can be accepted in the current version with some minor adjustment before publication.
Some points in current form of the manuscript should be corrected or detailed as following:
According to the Toxins template, the abstract should be as stated ‘A single paragraph of about 200 words maximum’, therefore it must be shortened.
Lines e.g. 12, 15, 80 – inconsistency: if speaking of all tested fumonisins it should be written in one, same, plural form in the text, same for aflatoxins. Correction should be made.
Line 16 and others – if the combined result of T2 and HT is placed than ‘/’ shouldn’t be used as it may indicates on result of T2 “or” HT2. Furthermore inconsistency in writing of HT2 and T2 also found as HT-2 and T-2. It should be unified.
Line 17 - ‘and deoxynivalenol was detected at a low level of 510 ppb in one sample.’ improve the style of this part of the sentence writing.
Table 1. - In my opinion it should be placed in the main text near to the first time it is cited, for example after the paragraph in which it is cited for the first time.
Paragraph 4 and 5 in Introduction section could be combined, as the data concern same subject.
Figure 1. - I do not find it relevant enough for showing in main paper, maybe it could be moved to the supplementary files?
General - The style of a few of the sentences need to be improved.
In my opinion the corrected manuscript would be worth publishing in Toxins.
Author Response
We sincerely appreciate your comments and we have made efforts to address the issues raised. We believe your input has enabled us to improve on the quality of this manuscript. Our responses are shown below.
According to the Toxins template, the abstract should be as stated ‘A single paragraph of about 200 words maximum’, therefore it must be shortened.
Action: we have reduced the number of words to 200.
Lines e.g. 12, 15, 80 – inconsistency: if speaking of all tested fumonisins it should be written in one, same, plural form in the text, same for aflatoxins. Correction should be made. This correction has been made.
Line 16 and others – if the combined result of T2 and HT is placed than ‘/’ shouldn’t be used as it may indicates on result of T2 “or” HT2. Furthermore inconsistency in writing of HT2 and T2 also found as HT-2 and T-2. It should be unified. We have unified this.
Line 17 - ‘and deoxynivalenol was detected at a low level of 510 ppb in one sample.’ improve the style of this part of the sentence writing.
We have rephrased this sentence to enhance clarity.
Table 1. - In my opinion it should be placed in the main text near to the first time it is cited, for example after the paragraph in which it is cited for the first time.
Action: we have moved this so that it appears immediately after the introduction.
Paragraph 4 and 5 in Introduction section could be combined, as the data concern same subject.
Action: we have merged these two paragraphs and edited to enhanced coherency in the content.
Figure 1. - I do not find it relevant enough for showing in main paper, maybe it could be moved to the supplementary files?
Action: Given that this manuscript will be read by people of diverse origin, including those that do not know the country and the location, we would like to keep this figure in the current section of the manuscript.
General - The style of a few of the sentences need to be improved.
Thanks for this valuable comment. We have gone through the document and made changes in the writing style and grammar. We hope the current version of the manuscript would be clearer to the readers.
Issue 1. Paragraph 4 and 5 in Introduction section could be combined, as the data concern same subject.
Action: we have merged these two paragraphs.